# Prevalence of pulmonary tuberculosis among casual labourers working in selected road construction sites in central Uganda

Ivan Ahimbisibwe[1,2]*, Cathbert Tumusiime[1,3], Laban Muteebwa[1], Ezekiel Mupere[4], Irene Andia Biraro[5,6]

1 Clinical Epidemiology Unit, School of Medicine, Makerere University-College of Health Sciences, Kampala, Uganda, 2 Department of Roads and Bridges, Ministry of Works and Transport, Kampala, Uganda, 3 Department of Programs, Think Well Institute, Kampala, Uganda, 4 Department of Paediatrics and Child Health, School of Medicine, Makerere University, College of Health Sciences, Kampala, Uganda, 5 Department of Internal Medicine, School of Medicine, Makerere University, College of Health Sciences, Kampala, Uganda, 6 Uganda Virus Research Institute/Medical Research Council, Entebbe, Uganda

* aivan9850@gmail.com

## Abstract

### Introduction

Workers with occupational exposure to respirable silica dust, such as casual labourers at road construction sites (RCSs), are known to be at high risk of developing pulmonary tuberculosis (TB). There is limited literature about the burden of PTB among this subpopulation with high occupational exposure to silica dust at road construction sites. We aimed to determine the prevalence of PTB among casual labourers working at road construction sites in central Uganda.

### Methods

We enrolled 297 participants via consecutive sampling in a cross-sectional study between September 1st and September 30th, 2022, at four road construction sites in four districts in central Uganda. A structured questionnaire was administered, and the PTB patients were identified by using GeneXpert and/or computer-aided detection for TB (CAD4TB). The data were analysed with STATA version 17.0. Descriptive statistics adjusted for clustering were used to summarize the data, and the relationships between PTB and independent variables were assessed by using a mixed effects modified Poisson regression model to estimate the adjusted prevalence ratios.

### Results

Most participants were males (95.6% [284/297]), and the median age was 29 years (interquartile range [IQR]: 25–33). The prevalence of PTB among casual labourers was 2.4% (95% CI: 1.9, 2.8). Not being vaccinated with BCG (3.45, 95% CI: 1.02, 11.61), alcohol use (2.70, 95% CI: 1.52, 4.80) and staying in shared rooms (8.13, 95% CI: 4.37, 15.12) were positively associated with having PTB.

**Data Availability Statement:** The data generated from this study can be publicly accessed in dryad via this link: https://datadryad.org/stash/dataset/doi:10.5061/dryad.t4b8gtj8s and its unique digital

object identifier (DOI): doi:10.5061/dryad.t4b8gtj8s.

**Funding:** The author(s) received no specific funding for this work.

**Competing interests:** The authors have declared that no competing interests exist.

**Abbreviations:** ACF, Active Case Finding; AIDS, Acquire Immune Deficiency Syndrome; CAD4TB, Computer-aided detection for tuberculosis; COVID-19, Coronavirus Disease 2019; CXR, Chest X-ray; DLGs, District Local Governments; EPTB, Extrapulmonary Tuberculosis; HBCs, High Burden Countries; HIV, Human Immuno-Deficiency Virus; ICOH, International Commission on Occupational Health; ILO, International Labour Organization; ISTC, International Standards for Tuberculosis Care; IQR, Interquartile range; KCCA, Kampala Capital City Authority; LCS, Low-Cost Sealing; Mak, Makerere University; MoGLSD, Ministry of Gender, Labour and Social Development; MoH, Ministry of Health; MoWT, Ministry of Works and Transport; NTLP, National Tuberculosis & Leprosy Program; PI, Principal Investigator; PPE, Personal Protective Equipment; PTB, Pulmonary Tuberculosis; RCSs, Road construction sites; SARS-CoV-2, Severe Acute Respiratory Syndrome Coronavirus 2; SoMREC, School of Medicine Research Ethics; TB, Tuberculosis; TPP, Target Product Profile; UN, United Nations; UNRA, Uganda National Roads Authority; WHO, World Health Organization.

## Conclusion

There is a high prevalence of PTB among casual labourers working at road construction sites in central Uganda. Individuals who had never been vaccinated with BCG, alcohol users and those staying in shared rooms were at an increased risk of having PTB. We recommend routine screening of casual labourers at road construction sites to optimize active TB case finding.

## Background

An estimated 10.6 million people were diagnosed with tuberculosis (TB) worldwide in 2022, whereas in the same year, a total estimate of 1.3 million people died from TB [2]. According to the World Health Organization (WHO), for the distribution of the global TB burden for regions in 2022, Africa ranked second after Southeast Asia, with a total of 23% of the global TB cases [2].

Despite efforts by several stakeholders to stop TB, Uganda remains one of the 30 WHO-designated high-burden TB countries, with an incidence rate of 198 per 100,000 people [1,2]. High burden countries (HBCs) accounted for 87% of all of the estimated incidence of TB worldwide in 2022, as was the case in 2021 [2].

Notably, Uganda achieved 100% TB treatment coverage despite recording a total of 12,000 TB-related deaths in 2022 [2].

In 2018, the United Nations (UN) General Assembly designated casual labourers working at road construction sites as high-risk TB subpopulations due to their high occupational exposure to dust, especially from silica-containing surfaces [3]. Therefore, there is an urgent need to understand the burden of TB in this subpopulation and modifiable factors that can be leveraged to enhance prevention and control programs. Several studies have also reported that workers exposed to silica dust, such as miners, construction workers, and stone crushers (among others), are at increased risk of developing tuberculosis [4–6].

For example, miners have been reported to have a TB incidence ranging between 2.8% and 12.4% [7–11]. Casual labourers involved in the crushing and grading of aggregates from hard cores (rocks), excavation of gravel (murram) and sorting of gravel (murram), among others, are exposed to high amounts of silica dust [11,12].

The inhalation of respirable silica leads to silicosis, which is mainly characterized by granulomas with collagen nuclei surrounded by epithelioid cells and results in the formation of silicotic nodules that are dispersed throughout the lungs; as the condition worsens, these nodules can consolidate to form enormous masses to distort the parenchyma, thus leading to the development of many pulmonary diseases, including silica-associated PTB [13].

Individuals exposed to silica have a 2.8 to 3.9 times greater risk of developing pulmonary tuberculosis and a 3.7 times greater risk of developing extrapulmonary tuberculosis than does the general population [14]. Recent studies in Zimbabwe and India have shown that the prevalence of TB among workers exposed to silica dust (miners) ranges between 4% and 9.4%, which is relatively high [8,9].

Several studies in Uganda have focused on the burden of TB among other risk groups, such as health workers and slum dwellers [15–17]. However, no study has been conducted among road construction workers in Uganda, despite the high occupational exposure and widespread use of silica-containing surfaces.

Some Ugandan surfaces are made of sedimentary and asphaltic rocks and murram, which have been found to contain high amounts of respirable silica dust [3,6,18]; however, casual labourers often lack personal protective equipment and have long periods of exposure, which exposes them to high amounts of silica dust [19,20]. In addition, these casual labourers reside in poorly ventilated houses near construction sites and often stay and work as a group, which leads to overcrowding that enhances the transmission of PTB. Overcrowding was found to be associated with TB disease in a study among gold miners with the same occupational exposure in South Africa [21].

In addition, casual labourers have temporal migratory tendencies as they move from their homes to sites and later switch from one site to another. Recent studies have shown that geographical mobility and migration are associated with TB [10,22]. Based on these data, we aimed to determine the prevalence of PTB and associated factors among casual labourers working at road construction sites in central Uganda.

## Methods

### Study design and setting

This was a cross-sectional study conducted among four selected road construction project sites, with each site located in one of the four study districts in central Uganda. These were purposively selected because they were the only active project sites at the time of data collection, given that most of the projects were halted for a number of reasons, including shortfalls in financing and heavy rains, among others. The RCSs that participated in the study included the Busega-Mpigi Expressway (Mpigi), Kira-Matugga Road (Wakiso), Bamugolodde-Mayirit-Busansula Road (Nakasongola), and Bulwandi-Gyirira Swamp Crossway (Kayunga).

### Study population and eligibility criteria

We recruited 297 casual labourers working at road construction sites in four districts in central Uganda from September 1st to September 30th of 2022. Participants were eligible for inclusion if they were ≥18-years-old, had worked at the site for more than six months, were deployed at site stations as casual labourers and provided written informed consent.

Participants who were not available at the site during the period of data collection were excluded from the study.

### Sampling procedure and data collection

The study participants were found at selected road construction sites, and those who fulfilled the eligibility criteria were recruited for the study on a rolling basis until the required sample size per site was reached (consecutive sampling). The sample size was proportional to the number of casual labourers at each road construction site.

Enrolment per site was as follows: 250 out of 547 casual labourers at Busega-Mpigi Expressway, 24 out of 54 casual labourers at Kira-Matugga Road, 13 out of 28 casual labourers at Bamugolodde-Mayirit-Busansula Road and 10 out of 21 casual labourers at Bulwandi-Gyirira Swamp Crossway.

At the site, a semistructured questionnaire was administered to each of the eligible participants by a trained research assistant. The questionnaire included questions about sociodemographic, behavioural and clinical characteristics. The study participant would then proceed to the mobile clinic vehicle mounted with a CAD4TB chest X-ray machine for assessment and hand in both the completed questionnaire and the sputum container for those who managed

to produce the sputum. The procedure continued every day from 7 am to 9 am until the required sample size per site was obtained.

## TB screening

In this study, PTB was defined as either bacteriologically confirmed positive sputum from GeneXpert® MTB/RIF or a clinically confirmed diagnosis (a CAD4TB abnormal chest X-ray suggestive of TB with any TB symptoms).

Xpert MTB/RIF is a relatively new novel, integrated, cartridge-based molecular test that has been designed to test TB DNA that is highly specific for MTB. This test can also reveal a mutation that confers resistance to the most effective TB drug (rifampicin) to MTB. Xpert MTB/Rif is an automated test that was directly utilized in the central laboratory at Kampala on each of the 25 sputum samples as they were collected from the participants each day.

The artificial intelligence (AI)-fit CAD4TB system manufactured by Delft Imaging meets the target product profile (TPP) of triage tests of the WHO (sensitivity ≥90% and specificity≥70%); therefore, the WHO strongly recommends its use in high-risk populations [23]. CAD4TB considers a threshold score of 50 as being abnormal and suggestive of TB.

## Study variables

For the dependent variables, this study considered the presence or absence of PTB disease as a binary outcome. The presence of PTB was confirmed by either a positive sputum GeneXpert result or an abnormal chest X-ray in the presence of any TB symptoms. The absence of PTB was confirmed by a GeneXpert negative sputum result or a normal chest X-ray in the absence of TB symptoms.

The following independent variables were examined: age, sex, education level, marital status, role/position and site location. Age was recorded in years; sex was recorded as either female or male. Educational level was recorded as certificate (seven years of education), diploma (eleven years of education), bachelor's degree (fifteen years of education), or no formal education. Marital status was recorded as single, married, divorced or widowed. Site location was recorded as either urban or rural per district. Behavioural/lifestyle factors included alcohol use and a history of smoking.

Clinical factors included; HIV status, blood pressure, TB symptoms (coughing, weight loss, night sweats, chest pain, difficulty in breathing, and fever), history of comorbidities (diabetes mellitus, hypertension, asthma, chronic liver disease, cancer, and cardiac disease), TB treatment history, history of contact and the presence or absence of a BCG scar, and BMI.

## Data analysis

Data analysis was performed in STATA 17.0 (Texas, USA).

Continuous variables were summarized by using medians and interquartile ranges (IQRs), and categorical variables were summarized by using frequencies and percentages, which were all adjusted for clustering by study site.

Mixed effects modified Poisson regression was used to determine the factors associated with PTB. In the bivariate analysis, crude prevalence ratios (cPRs) and their corresponding 95% confidence intervals (CIs) were computed. Independent variables with a P value < 0.3 were considered for the multivariate analysis. In the multivariate analysis, the chi-square test was used to assess any possible interactions between independent variables. Confounding was assessed, and an independent variable that changed the prevalence ratio of another by a magnitude of 10% was considered to be a confounder. The adjusted prevalence ratios (aPRs), their corresponding CIs and P values in the final model are presented.

## Ethical considerations

The study protocol was approved by the Makerere University School of Medicine Research Ethics Committee (Mak-SOMREC) under approval number Mak-SOMREC-2022-388. Administrative approval was obtained from the Uganda National Roads Authority (UNRA) and the Ministry of Works and Transport (MoWT). Written informed consent was obtained from all of the study participants after providing all of the study information. The participants who were diagnosed with PTB were linked to nearby TB care centres.

## Results

### Sociodemographic characteristics of the study participants

A total of 297 casual labourers participated in the study, representing a 100% response rate. The median (IQR) age of the participants was 29 years (25–33), and most participants were males (95.6%). Most of the participants were not married (66.7%), had a certificate-level education (81.5%) and had an exposure period of 0 to 1 year (51.7%). A few of the individuals were smokers (17.8%), and approximately one-quarter of the participants used dust masks (28.3%). More than half of the participants were nonalcohol users (51.2%). Most of the participants (64.0%) worked on sites in which the site setting was considered to be rural (Table 1). It should be noted that dust masks are contractually available from the employers.

### Clinical factors

Only 10.1% of the study participants had ever been tested for TB; moreover, 3.4% were experiencing wasting, and 17.2% were hypertensive. Over two-thirds of the participants had ever been tested for HIV (68.7%), and 83.5% had ever received a BCG vaccine. Of those who

**Table 1. Sociodemographic characteristics of the 297 study participants from selected road construction sites in central Uganda.**

| Variable | Category | Frequency n (%) |
|---|---|---|
| Age | Median (IQR) | 29 (25–33) |
| Sex | Female | 13 (4.4) |
|  | Male | 284 (95.6) |
| Marital status | Married | 99 (33.3) |
|  | Not Married | 198 (66.7) |
| Education level | None | 33 (11.1) |
|  | Certificate | 242 (81.5) |
|  | Diploma or more | 22 (7.4) |
| Exposure period | 0 to 1year | 156 (52.5) |
|  | More than 1 year | 141 (47.5) |
| Smoking Status | No | 244 (82.2) |
|  | Yes | 53 (17.8) |
| Alcohol use | No | 152 (51.2) |
|  | Yes | 145 (48.8) |
| Use of dust mask | No | 213 (71.7) |
|  | Yes | 84 (28.3) |
| Site Setting | Urban | 107 (36.0) |
|  | Rural | 190 (64.0) |

IQR-Interquartile Range.

**Table 2. Clinical characteristics of the 297 study participants from selected road construction sites in central Uganda.**

| Variable | Category | Frequency n (%) |
|---|---|---|
| BMI | No wasting | 287 (96.6) |
| | Wasting | 10 (3.4) |
| Hypertension | No | 246 (82.8) |
| | Yes | 51 (17.2) |
| Ever tested for TB | No | 267 (89.9) |
| | Yes | 30 (10.1) |
| Previous TB history* | Negative | 25 (83.3) |
| | Positive | 4 (13.3) |
| | Do not remember | 1 (3.33) |
| HIV status | Negative | 196 (66.0) |
| | Positive | 7 (2.4) |
| | Do not remember | 94 (31.6) |
| BCG scar | No | 49 (16.5) |
| | Yes | 248 (83.5) |

*n = 30.

had ever been tested for TB, only 13.3% had positive test results. Of those who had ever tested for HIV, only 2.9% tested positive (Table 2).

The study used the standard definition of the WHO of wasting in participants as having a low weight for a particular height.

## Prevalence of TB

In this study, the prevalence of PTB was 2.4% (95% CI: 1.9–2.8), and all of the patients were males.

## Bivariate and multivariate analyses of the factors associated with TB

Room occupancy, alcohol use, BMI, BCG scarring, exposure period, smoking status, marital status and age were the only factors that were considered for the multivariate analysis, because they had P values less than 0.3. After adjusting for covariates, only alcohol use (P = 0.001), staying in shared rooms (P<0.001), and absence of a BCG scar (P = 0.046) were found to be significantly associated with PTB (Table 3). No interactions or confounders were found in the final model.

From Table 3, the only factors that were significantly associated with PTB among the casual labourers working in selected road construction sites in central Uganda (according to the multivariate analysis) were sharing rooms of residence (aPR = 8.133, 95% CI; 4.374, 15.123), alcohol use (aPR = 2.700, 95% CI; 1.521, 4.792) and absence of a BCG scar (aPR = 3.447, 95% CI; 1.024, 11.606).

## Discussion

This study assessed the prevalence of PTB among casual labourers from selected road construction sites in central Uganda. In this cross-sectional study, the TB incidence was found to be 10 times greater than the national TB incidence (0.2%) in Uganda [24]. This observation can be attributed to the fact that this study concentrated on only a high-risk group of silica dust-exposed workers (casual labourers) involved in road work, whereas the coverage of the

**Table 3. Bivariate multivariate analyses of the factors associated with PTB.**

| Variable | Prevalence of PTB n (%) | Bivariate analysis | | Multivariate analysis | |
|---|---|---|---|---|---|
| | | cPR (95% CI) | P Value | aPR (95% CI) | P Value |
| **Age** | | 1.04 (0.98,1.10) | **0.234** | | |
| **Marital status** | | | | | |
| Not Married | 2 (2.0) | Ref | | | |
| Married | 5 (2.5) | 1.25 (0.87, 1.80) | **0.231** | | |
| **Exposure period** | | | | | |
| 0–1 year | 3 (1.9) | Ref | | | |
| Above 1 year | 4 (2.8) | 1.48 (0.97, 2.25) | **0.072** | | |
| **Room occupancy** | | | | | |
| One person | 1 (0.6) | Ref | | Ref | |
| More than person | 6 (5.0) | 8.85 (4.83, 16.21) | **<0.001** | 8.13 (4.37,15.12) | **<0.001** |
| **Site Setting** | | | | | |
| Rural | 5 (2.6) | 1.41 (0.56, 3.55) | 0.468 | | |
| Urban | 2 (1.9) | Ref | | | |
| **Wearing dust mask** | | | | | |
| No | 5 (2.4) | Ref | | | |
| Yes | 2 (2.4) | 1.01 (0.38, 2.71) | 0.978 | | |
| **BMI** | | | | | |
| No wasting | 6 (2.0) | Ref | | | |
| Wasting | 1 (10.0) | 4.78 (2.54,9.01) | | | |
| **Smoking status** | | | | | |
| Nonsmokers | 5 (2.1) | Ref | | | |
| Smokers | 2 (3.7) | 1.8 (0.706, 4.59) | | | |
| **Alcohol use** | | | | | |
| Yes | 2 (1.3) | 2.62 (1.687,4.071) | | 2.70 (1.52,4.80) | **0.001** |
| No | 5 (3.4) | Ref | | Ref | |
| **Having at least one symptom** | | | | | |
| Yes | 4 (2.4) | 0.97 (0.42,2.22) | | | |
| No | 3 (2.3) | Ref | | | |
| **BCG Vaccination status** | | | | | |
| Yes | 4 (1.6) | Ref | | Ref | |
| No | 3 (6.1) | 3.80 (1.17,12.28) | | 3.45 (1.02,11.61) | **0.046** |

national TB prevalence survey 2015 [25] included other low-risk groups. Although there are no studies on the prevalence of TB among silica dust-exposed workers in Uganda, a study in Ghana found that the TB prevalence among miners (silica dust-exposed) was 2.65%, whereas it was 0.64% among nonminers who were screened in the community (Ohene et al., 2021). Another study among 451 small-scale artisanal miners who were also silica-dust-exposed workers in Zimbabwe reported a TB incidence of 4% (Moyo et al., 2021). The difference in incidence could be due to a number of reasons, such as greater precision given the large sample size that was used in the later study. In 2017, the Centers for Disease Control reported that the prevalence of TB among South African miners is 15 times greater than that among the rest of the South African population [26].

This high prevalence of TB among casual labourers involved in road work implies that road construction workers could have a considerable burden of infectious TB disease, which calls for urgent attention and implementation of TB prevention and control measures among road construction workers.

The study participants who reported of sleeping in shared rooms of residence were 8.1 times more likely to have PTB than those who did not share their rooms. A more recent study in Uganda by Kirenga et al. showed that overcrowding was a strong risk factor associated with TB [17]. This finding concurs with the findings of several other studies evaluating environmental risk factors associated with TB [27,28].

It was also found that study participants who drank alcohol were 2.7 times more likely to have PTB than those who did not drink alcohol. Several other studies by Lienhardt [29], among others, have also demonstrated alcohol use to be among the top modifiable risk factors associated with TB [30–33]. This can be attributed to the fact that most bars are congested, and those drinking alcohol are consistently in close contact, which puts them at risk of contracting TB if there is a positive TB case. The WHO global TB report of 2022 also highlighted alcohol use as being among the five risk factors that are strongly associated with many TB cases [1].

In this study, participants without a BCG scar were 3.4 times more likely to have PTB than those with a BCG scar. This finding could be best explained by the absence of immunity against TB disease on the assumption that the absence of a BCG scar implies that a particular participant was either not vaccinated with BCG at birth or may not have elicited an immune response when vaccinated. This finding is also consistent with a study in the United States that demonstrated BCG vaccination to be protective against TB disease [34]. Some countries experiencing high TB incidence have resorted to targeted BCG vaccination in adults younger than 35 years of age among high-risk groups [6,35–37].

Two strengths of this study were the fact that an updated version (7.0) of the CAD4TB software with greater sensitivity (91.5%) was used in the study and that the study was the first to be performed among silica dust-exposed workers in Uganda, particularly casual labourers working at road construction sites who represent a high-risk group. Notably, all casual labourers who were approached consented voluntarily, thus implying that the study achieved a 100% response rate.

This study had limitations that should be noted. Information bias could be detected in the way that HIV status (which is a known predictor of TB disease) may have been underreported by some of the participants, due to the fact that the study did not perform HIV tests but rather used self-reports. Due to the fact that some study participants failed to produce sputum, there may have been under declaration of TB in this study. The study was prone to random error in such a way that the sample size was not adequate to examine all of the associated factors. As a result, the study lacked enough power to detect small effects. This study may not be generalizable to the entire country, such as Uganda, due to ecological differences in earth surfaces that contain different amounts of silicon, thus implying varying exposure levels for different regions in the country.

## Conclusions

There was a high prevalence of TB among casual labourers working at road construction sites in central Uganda. Study participants living in shared rooms had greater odds of contracting TB than did those sleeping one person per room. TB incidence was also greater among alcohol users than among nonusers. Participants without a BCG scar had relatively greater odds of developing PTB than those with a BCG scar. Routine TB screening of road construction workers at all road construction sites is recommended, and targeted BCG vaccination in road

construction workers younger than 35 years could be adopted to increase immunity against TB disease. A more comprehensive study covering all regions in the country should be conducted to ascertain the overall TB incidence among road construction workers.

## Acknowledgments

We wish to extend our gratitude to the research assistants, Uganda National Roads Authority, Ministry of Works and Transport, and National Tuberculosis and Leprosy Program under the Ministry of Health, USAID, The Global Fund, all who made this research possible. We also wish to thank the project managers of all of the selected road construction projects that participated in this research.

We thank American Journal Experts (AJE) for English language editing.

## Author Contributions

**Conceptualization:** Ivan Ahimbisibwe, Ezekiel Mupere, Irene Andia Biraro.

**Data curation:** Ivan Ahimbisibwe, Cathbert Tumusiime, Laban Muteebwa.

**Formal analysis:** Ivan Ahimbisibwe, Cathbert Tumusiime, Laban Muteebwa.

**Methodology:** Ivan Ahimbisibwe, Cathbert Tumusiime, Laban Muteebwa.

**Project administration:** Ivan Ahimbisibwe.

**Resources:** Ivan Ahimbisibwe.

**Software:** Ivan Ahimbisibwe.

**Supervision:** Ezekiel Mupere, Irene Andia Biraro.

**Validation:** Ivan Ahimbisibwe.

**Writing – original draft:** Ivan Ahimbisibwe.

**Writing – review & editing:** Ivan Ahimbisibwe, Cathbert Tumusiime, Laban Muteebwa, Ezekiel Mupere, Irene Andia Biraro.

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
