## [Decision Letter · Decision Letter 0]

2 Apr 2024

PONE-D-24-03066PREVALENCE OF PULMONARY TUBERCULOSIS AMONG CASUAL LABOURERS WORKING IN SELECTED ROAD CONSTRUCTION SITES IN CENTRAL UGANDAPLOS ONE

Dear Dr. AHIMBISIBWE,

Thank you for submitting your manuscript to PLOS ONE. After careful consideration, we feel that it has merit but does not fully meet PLOS ONE’s publication criteria as it currently stands. Therefore, we invite you to submit a revised version of the manuscript that addresses the points raised during the review process.

We look forward to receiving your revised manuscript.

Kind regards,

Stephen Michael Graham, FRACP, PhD

Academic Editor

PLOS ONE

A clean copy of the edited manuscript (uploaded as the new *manuscript* file).

Reviewers' comments:

Reviewer's Responses to Questions

**Comments to the Author**

1. Is the manuscript technically sound, and do the data support the conclusions?

Reviewer #1: Yes

Reviewer #2: Partly

2. Has the statistical analysis been performed appropriately and rigorously? 

Reviewer #1: Yes

Reviewer #2: Yes

3. Have the authors made all data underlying the findings in their manuscript fully available?

Reviewer #1: Yes

Reviewer #2: No

4. Is the manuscript presented in an intelligible fashion and written in standard English?

Reviewer #1: Yes

Reviewer #2: Yes

5. Review Comments to the Author

Reviewer #1: The authors present important data on the TB prevalence among a non – prioritized group (casual laborers at road construction sites) in the Uganda TB response. Informative study for other NTPs.

Abstract:

-Results

- The authors indicate that being BCG vaccinated was positively associated with TB contrary to what they include in the conclusion as well as then main results & discussion section. The info should be consistent.

-Background

- There is a more recent global TB report (2023) that the authors can make reference to.

- Sentence structure in some areas within the background can be improved…. For example, the authors state that “Exposure to silica has a 2.8 to 3.9 times higher risk of developing pulmonary tuberculosis and a 3.7 times higher risk of developing extra-pulmonary tuberculosis than the general population”

- study design, setting

- The length of the first sentence impairs its clarity

-Sampling procedure and data collection

- The authors state that a participant with a positive result on either or both of the tests was taken as a PTB case. Does this imply that CAD4TB was used for diagnosis without bacteriological confirmation? The authors do not list the threshold CAD4TB score that was used for this study. It is possible that not all participants with a positive CAD4TB had TB especially in this population where other lung conditions are prevalent. This may be reflected as a study limitation if sputum was not tested for some participants.

- There is repetition on CAD4TB & artificial intelligence

- Clinical factors

- The authors do not list a definition for wasting under the study methods

- Inconsistency with BCG scar vs no BCG scar – The authors state that BCG scar was significantly associated with TB.

- study limitations

- The early morning sputum sample has traditionally been known to have a better yield vs what the authors state.

-Conclusions

- The authors recommend targeted BCG vaccination for road construction workers – no evidence for this is listed. Additionally, BCG is more effective in preventing complicated TB forms.

Reviewer #2: This article describes a study assessing the prevalence of pulmonary TB among casual labourers working in selected road construction sites in central Uganda. It seems to arise from a Masters thesis (http://dspace.mak.ac.ug/handle/10570/11586). It has the potential to add to the evidence base as there is no published work focused on this population. The methodology appears likely to be appropriate, however there are multiple gaps in the paper as currently written such that it is not suitable for publication in its current form.

Major issues to address:

1. Participants and sampling: The participants were from the only RCS that were active at the time of data collection. Participant numbers were proportional to workforce size and included consecutively during recruitment period each day. The representativeness per site and compared to other usually active RCS in central Uganda are key to whether the results are valid and useful.

a. No information is provided about the 4 RCS (other than name/location) to allow the reader to know how representative these sites may be of other sites that were not currently active – might there have been substantial difference in workers at these sites compared to others? Could potential workforce size at these sites increased as workers came from inactive sites to seek work?

b. The total potential workforce sizes (understanding they were casual labourers) and the proportion of each workforce included are not provided. Did all workers arrive on site at the same time, and all have an equal chance of participating?

2. Results should include detail on the screening assessments, not just the final prevalence. What proportions were positive on symptom screen, chest x-ray and Xpert MTB, alone and in combination? This will help readers involved in screening activities interpret how this cohort may be similar or different to their own work.

3. No information about actual silica exposure at RCS is provided. The largest point estimate for risk factors was living in crowded accommodation, which might occur with forms of work that are not related to road construction and have no exposure to silica dust. The focus on silica dust in the introduction and discussion is not well linked to the actual results provided.

Other issues to address:

1. Data analysis methods state that descriptive results are adjusted for clustering by study sites, but frequency and percentages are given as totals across all sites and it is not clear how or why these values would be adjusted for clustering. Was adjustment for clustering done in the bivariable and multivariable analysis rather than the descriptive analysis?

2. Results state that 100% of approached participants consented. This is surprisingly high and raises the concern that participation was not truly voluntary. However, it may be that the screening offered was understood and valued by the participants. If truly informed voluntary consent then this is a strength and a brief comment in the discussion on the participation rate would be worthwhile.

3. More detail is needed on the screening. What did the symptom screen include? Which GeneXpert test used – was it all Xpert MTB/RIF? What was the threshold for a positive score with CAD4TB? Both tests require details provided about the manufacturer. The limitations notes an issue with poor quality sputum, so reporting the number of people who produced a sample and the proportion that could be tested would be useful.

4. How were various socio-demographic characteristics defined?

a. Can you describe ‘certificate level’ education in a way that helps international readers – how many years of schooling would this equate to?

b. Is ‘exposure period’ the duration of working on road construction?

c. Alcohol use it also referred to in a table as ‘alcoholic’ which has a high alcohol use implication – how was alcohol use defined (e.g. ever/never or a certain number of units of alcohol per week)?

d. How was smoking defined (e.g. ever/never or current/past/never or a certain number of cigarettes or tobacco equivalent per week?).

e. It would be helpful to specify if dust masks were made universally available by employers or needed to be provided by the casual labourer.

f. How is a ‘chronic illness’ defined? How does that relate to being hypertensive?

5. Table 7 is referred to in the text but it should be Table 3. The table refers to PR but the text refers to IRR. The abstract is not clear what is being reported in results of association. The abstract says that being vaccinated with BCG is a risk whereas the text specifies that it is not being vaccinated which is a risk.

6. The text is intelligible, but the introduction would benefit from substantial revision, especially where some of the references are not relevant to the text they are attached to. Attributions in the discussion are at times overstated.

7. A conclusion stating that all participants with PTB were male is not surprising given 95.6% of participants were male. If the proportion of participants who were male is not representative of road construction workers throughout central Uganda then that may deserve comment.

8. Data availability statement agrees to make data available from the corresponding author on reasonable request. The Plos One data availability FAQ indicate that a single author as point of contact is insufficient.

6. PLOS authors have the option to publish the peer review history of their article (what does this mean?). If published, this will include your full peer review and any attached files.

Reviewer #1: No

Reviewer #2: No

---

## [Author Response · Author response to Decision Letter 0]

16 May 2024

Ahimbisibwe Ivan

C/o Makerere University -Clinical Epidemiology Unit.

avian9850@gmail.com

+256-773110519

17th April 2024

The Academic Editor 

PLOS ONE Tuberculosis

RE: Submission of correspondences to issues raised on my submitted manuscript

The above subject matter refers; This is to provide response to a number of issues raised on my manuscript that I submitted to you for publishing “Prevalence of Pulmonary Tuberculosis among casual labourers working in selected road construction sites in Central Uganda”. 

A) Academic Editor

Answer: Yes, the manuscript has been presented according to PLOS ONE’s style requirements.

Answer: The manuscript was edited by AJE as advised.

Answer: The data has been deposited in dryad a scientific data repository on the following doi:10.5061/dryad.t4b8gtj8s. 

NB: All changes referred to herein are in the manuscript saved with tracked changes.

Answer: Yes, the ethics statement is only under the methods section, having been deleted from the acknowledgements section as may be seen from line 428 to 443.

B) Reviewer 1

Abstract:

-Results

- The authors indicate that being BCG vaccinated was positively associated with TB contrary to what they include in the conclusion as well as then main results & discussion section. The info should be consistent.

Answer: This has been rectified in the abstract as can be seen from line 42 to 44, to read “Not being vaccinated with BCG (3.45, 95% CI (1.02, 11.61)), alcohol use (2.70, 95% CI (1.52, 4.80)) and staying in shared rooms (8.13, 95% CI (4.37, 15.12)) were positively associated with having PTB.”

-Background

- There is a more recent global TB report (2023) that the authors can make reference to.

Answer: This has been adopted as can be noticed in the manuscript from line 78 to 93.

- Sentence structure in some areas within the background can be improved…. For example, the authors state that “Exposure to silica has a 2.8 to 3.9 times higher risk of developing pulmonary tuberculosis and a 3.7 times higher risk of developing extra-pulmonary tuberculosis than the general population”

Answer: This has been clearly sorted for example the highlighted paragraph on line 110 to 112 now reads “ Individuals exposed to silica have a 2.8 to 3.9 times greater risk of developing pulmonary tuberculosis and a 3.7 times greater risk of developing extrapulmonary tuberculosis than does the general population.”

- study design, setting

- The length of the first sentence impairs its clarity

Answer: An improvement has been made as seen on line 135 to 140 to read as “This was a cross-sectional study conducted among four selected road construction project sites, each located in one of the four study districts in Central Uganda. These were purposively selected because they were the only active project sites at the time of data collection given that most of the projects were halted for a number of reasons, including shortfalls in financing and heavy rains, among others”.

-Sampling procedure and data collection

- The authors state that a participant with a positive result on either or both of the tests was taken as a PTB case. Does this imply that CAD4TB was used for diagnosis without bacteriological confirmation? The authors do not list the threshold CAD4TB score that was used for this study. It is possible that not all participants with a positive CAD4TB had TB especially in this population where other lung conditions are prevalent. This may be reflected as a study limitation if sputum was not tested for some participants.

Answer: This has been sorted as seen on lines 162 -177 and 340-342.

- There is repetition on CAD4TB & artificial intelligence

Answer: This has been cleared as can be seen on line 173-174.

- Clinical factors

- The authors do not list a definition for wasting under the study methods

Answer: A modest definition on wasting has been provided as seen on lines 246 – 247.

- Inconsistency with BCG scar vs no BCG scar – The authors state that BCG scar was significantly associated with TB.

Answer: This was an error and has been rectified as seen on lines 42 to 44.

- study limitations

- The early morning sputum sample has traditionally been known to have a better yield vs what the authors state.

Answer: The authors do not dispute this fact but acknowledge the existing challenge of study participant’s failure to produce sputum given casual labourers are working in dusty conditions.

-Conclusions

- The authors recommend targeted BCG vaccination for road construction workers – no evidence for this is listed. Additionally, BCG is more effective in preventing complicated TB forms.

Answer: There are recent studies which affirmed this and have been referenced on lines 326-328 specifically reference no. 35 and 38. The referenced studies state that there’s some level of efficacy in young adults and not fully effective.

C) Reviewer 2

Major issues to address:

1. Participants and sampling: The participants were from the only RCS that were active at the time of data collection. Participant numbers were proportional to workforce size and included consecutively during recruitment period each day. The representativeness per site and compared to other usually active RCS in central Uganda are key to whether the results are valid and useful.

a. No information is provided about the 4 RCS (other than name/location) to allow the reader to know how representative these sites may be of other sites that were not currently active – might there have been substantial difference in workers at these sites compared to others? Could potential workforce size at these sites increased as workers came from inactive sites to seek work?

Answer: We have made an amendment to the section sampling and data collection to reflect the numbers as may be seen on lines 154 to 157.

b. The total potential workforce sizes (understanding they were casual labourers) and the proportion of each workforce included are not provided. Did all workers arrive on site at the same time, and all have an equal chance of participating?

Answer: This has been catered for on lines 154 to 157. 

2. Results should include detail on the screening assessments, not just the final prevalence. What proportions were positive on symptom screen, chest x-ray and Xpert MTB, alone and in combination? This will help readers involved in screening activities interpret how this cohort may be similar or different to their own work.

Answer: This has been cleared as may be seen on lines 161 to 178.

3. No information about actual silica exposure at RCS is provided. The largest point estimate for risk factors was living in crowded accommodation, which might occur with forms of work that are not related to road construction and have no exposure to silica dust. The focus on silica dust in the introduction and discussion is not well linked to the actual results provided.

Answer: We used period of exposure variable to substitute actual volumes of silica dust exposure. Much as the variable was significant at bivariate level, it did not become significant at multivariate level as may be seen in Table 3 between lines 281 to 282.

Other issues to address:

1. Data analysis methods state that descriptive results are adjusted for clustering by study sites, but frequency and percentages are given as totals across all sites and it is not clear how or why these values would be adjusted for clustering. Was adjustment for clustering done in the bivariable and multivariable analysis rather than the descriptive analysis?

Answer: 

Adjusting for clustering only causes a change in the confidence intervals and not necessarily in the absolute measures such mean. 

2. Results state that 100% of approached participants consented. This is surprisingly high and raises the concern that participation was not truly voluntary. However, it may be that the screening offered was understood and valued by the participants. If truly informed voluntary consent then this is a strength and a brief comment in the discussion on the participation rate would be worthwhile.

Answer: Yes, all the participants consented especially after realising we were offering a free TB test. The strength has been included in the discussion as may be seen on lines 334-335.

3. More detail is needed on the screening. What did the symptom screen include? Which GeneXpert test used – was it all Xpert MTB/RIF? What was the threshold for a positive score with CAD4TB? Both tests require details provided about the manufacturer. The limitations notes an issue with poor quality sputum, so reporting the number of people who produced a sample and the proportion that could be tested would be useful.

Answer: 50 was the threshold for a positive score. Details about the manufacturer have been included as seen in lines 173 to 177.

4. How were various socio-demographic characteristics defined?

a. Can you describe ‘certificate level’ education in a way that helps international readers – how many years of schooling would this equate to?

Answer: Thank you, this has been rectified as may be seen on lines 190 to 191. 

b. Is ‘exposure period’ the duration of working on road construction?

Answer: Yes, time a casual labourer spends on road works.

c. Alcohol use it also referred to in a table as ‘alcoholic’ which has a high alcohol use implication – how was alcohol use defined (e.g. ever/never or a certain number of units of alcohol per week)?

Answer: Alcoholic status was an error and this has been changed to read alcohol use as seen in Table 1 between lines 238 to 239. 

d. How was smoking defined (e.g. ever/never or current/past/never or a certain number of cigarettes or tobacco equivalent per week?).

Answer: Smoking was defined as ever/never used cigarettes/tobacco.

e. It would be helpful to specify if dust masks were made universally available by employers or needed to be provided by the casual labourer.

Answer: It’s the responsibility of employers to avail personal protective gear including dust masks, however on all sites only a few casual labourers had the masks. This has been noted on lines 225 to 226.

f. How is a ‘chronic illness’ defined? How does that relate to being hypertensive?

Answer: chronic illnesses included diabetes, hypertension and HIV. Since these were presented separately, the “chronic illness” variable has been deleted as seen on line 244.

5. Table 7 is referred to in the text but it should be Table 3. The table refers to PR but the text refers to IRR. The abstract is not clear what is being reported in results of association. The abstract says that being vaccinated with BCG is a risk whereas the text specifies that it is not being vaccinated which is a risk.

Answer: Thank you, these errors have been corrected as can be seen between line 278. 

6. The text is intelligible, but the introduction would benefit from substantial revision, especially where some of the references are not relevant to the text they are attached to. Attributions in the discussion are at times overstated.

Answer: Thank you, the authors have made an adjustment.

7. A conclusion stating that all participants with PTB were male is not surprising given 95.6% of participants were male. If the proportion of participants who were male is not representative of road construction workers throughout central Uganda then that may deserve comment.

Answer: This is very helpful and taken into consideration as may observed on line 352.

8. Data availability statement agrees to make data available from the corresponding author on reasonable request. The Plos One data availability FAQ indicate that a single author as point of contact is insufficient.

Answer: Yes, the data is now available in a public depository called dryad on this doi: 10.5061/dryad.t4b8gtj8s.

---

## [Editor Report · Decision Letter 1]

17 May 2024

PREVALENCE OF PULMONARY TUBERCULOSIS AMONG CASUAL LABOURERS WORKING IN SELECTED ROAD CONSTRUCTION SITES IN CENTRAL UGANDA

PONE-D-24-03066R1

Dear Dr Ahimbisibwe,

We’re pleased to inform you that your manuscript has been judged scientifically suitable for publication and will be formally accepted for publication once it meets all outstanding technical requirements.

Kind regards,

Steve Graham

Stephen Michael Graham, FRACP, PhD

Academic Editor

PLOS ONE
---

## [Editor Report · Acceptance letter]

30 May 2024

PONE-D-24-03066R1 

PLOS ONE

Dear Dr. AHIMBISIBWE, 

I'm pleased to inform you that your manuscript has been deemed suitable for publication in PLOS ONE. Congratulations! Your manuscript is now being handed over to our production team.

Kind regards, 

on behalf of

Dr. Stephen Michael Graham 

Academic Editor

PLOS ONE